# Free-energy simulations reveal molecular mechanism for functional switch of a DNA helicase

Wen Ma[1,2,3,4], Kevin D Whitley[1,3], Yann R Chemla[1,4,3]*,
Zaida Luthey-Schulten[1,4,2,3,5]*, Klaus Schulten[1,4,2,3]†

[1]Center for the Physics of Living Cells, University of Illinois at Urbana-Champaign, Champaign, United States; [2]Beckman Institute for Advanced Science and Technology, Champaign, United States; [3]Center for Biophysics and Computational Biology, University of Illinois at Urbana-Champaign, Champaign, United States; [4]Department of Physics, University of Illinois at Urbana-Champaign, Champaign, United States; [5]Department of Chemistry, University of Illinois at Urbana-Champaign, Champaign, United States

**Abstract** Helicases play key roles in genome maintenance, yet it remains elusive how these enzymes change conformations and how transitions between different conformational states regulate nucleic acid reshaping. Here, we developed a computational technique combining structural bioinformatics approaches and atomic-level free-energy simulations to characterize how the *Escherichia coli* DNA repair enzyme UvrD changes its conformation at the fork junction to switch its function from unwinding to rezipping DNA. The lowest free-energy path shows that UvrD opens the interface between two domains, allowing the bound ssDNA to escape. The simulation results predict a key metastable 'tilted' state during ssDNA strand switching. By simulating FRET distributions with fluorophores attached to UvrD, we show that the new state is supported quantitatively by single-molecule measurements. The present study deciphers key elements for the 'hyper-helicase' behavior of a mutant and provides an effective framework to characterize directly structure-function relationships in molecular machines.
DOI: https://doi.org/10.7554/eLife.34186.001

*For correspondence:
ychemla@illinois.edu (YRC);
zan@illinois.edu (ZL-S)

†deceased

**Competing interests:** The authors declare that no competing interests exist.

## Introduction

Helicases are ubiquitous motor proteins that move along nucleic acids and separate duplex DNA or RNA into its component strands. This role is critical for various aspects of DNA and RNA metabolism; defects in helicase function in humans can lead to genomic instability and a predisposition to cancer (*van Brabant et al., 2000*; *Brosh, 2013*). Characterizing the atomistic mechanism for helicase function, although challenging, is crucial to link protein structure with their function and help engineering helicases with novel activities (*Arslan et al., 2015*).

DNA helicases can unwind double-stranded DNA (dsDNA) into single-stranded DNA (ssDNA), which are later copied during DNA replication or modified in DNA repair processes (*Wu and Spies, 2013*; *Lohman et al., 2008*). They are classified into six superfamilies (SF), among which SF1 and SF2 helicases are the largest superfamilies and share many similar conserved motifs. The minimal functional units for SF1 and SF2 helicases are monomers that contain two RecA-like motor domains for ATP hydrolysis (*Singleton et al., 2007*). SF1 helicases can unwind dsDNA by translocating on a ssDNA strand as shown in *Figure 1a*. Such translocation happens in a stepwise manner, during which the chemical energy from ATP hydrolysis is used to break the bonds in dsDNA via conformational changes of the motor domains (*Yang, 2010*; *Patel and Donmez, 2006*). An exemplary *Escherichia*

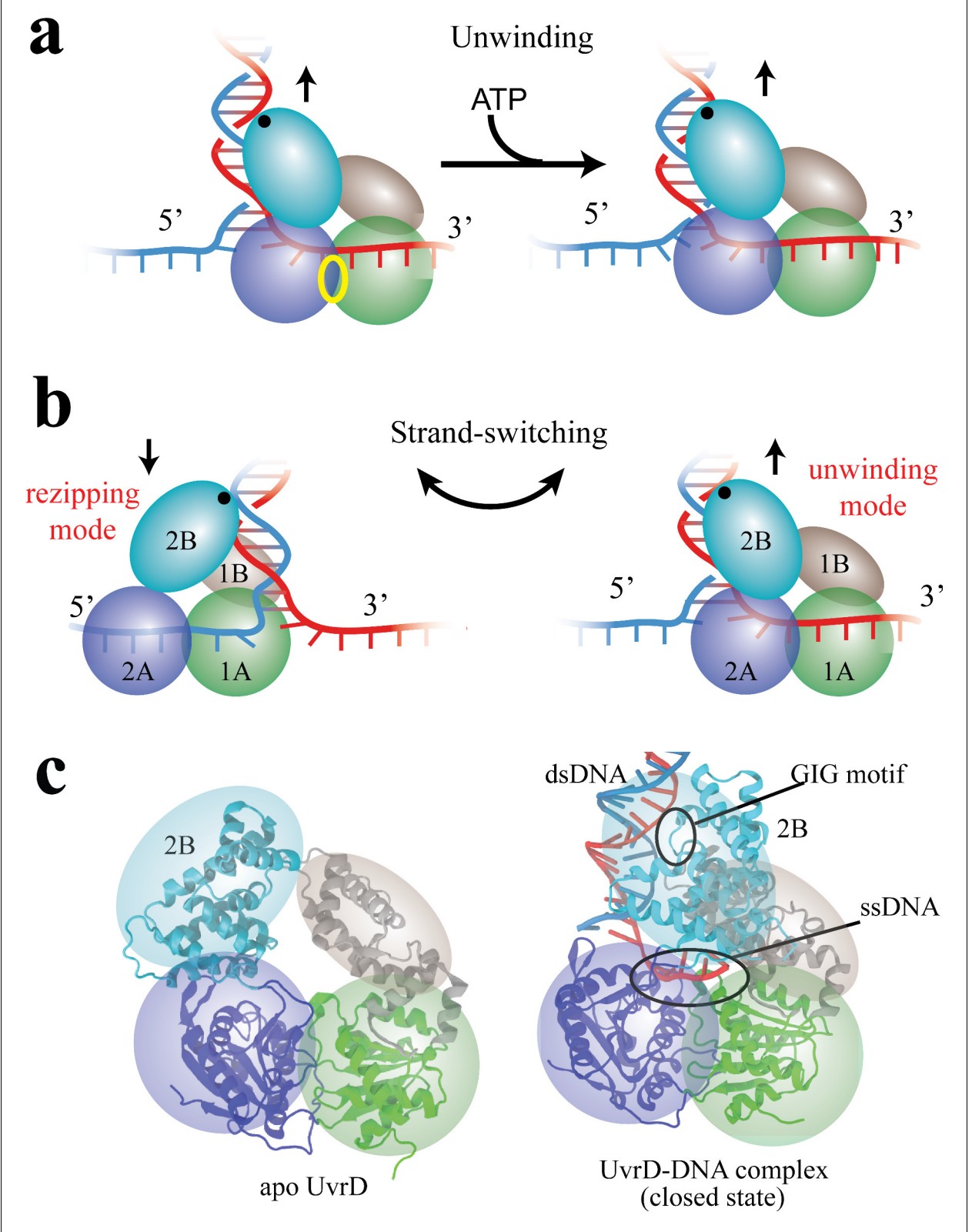

**Figure 1.** Functional switching of UvrD helicase. (a) Schematic illustration of unwinding by a DNA helicase (UvrD). The helicase uses chemical energy from ATP (hydrolyzed at the site labeled with a yellow ellipse between two motor domains) to unwind dsDNA. (b) A proposed model of UvrD functional switching at the fork junction. 1A, 2A, 1B and 2B domains are labeled in green, blue, gray and cyan, respectively. The UvrD conformation on the left

*Figure 1 continued on next page*

*Figure 1 continued*

represents the rezipping state whereas the conformation on the right represents the unwinding state. GIG motif (residues 414 to 422), drawn as a black dot, is important for UvrD interacting with dsDNA. (c) Crystal structures for apo UvrD (3LFU) and UvrD-DNA complex (2IS2, the closed state).
DOI: https://doi.org/10.7554/eLife.34186.002
The following figure supplement is available for figure 1:

**Figure supplement 1.** UvrD is found unlikely to adopt the apo conformation when bound to a fork junction.
DOI: https://doi.org/10.7554/eLife.34186.003

*coli* helicase, UvrD, belonging to SF1, has many cellular roles such as methyl-directed mismatch repair (*Iyer et al., 2006*; *Spies and Fishel, 2015*) and nucleotide excision repair (*Sancar, 1996*) by unwinding duplex DNA. UvrD can also prevent deleterious recombination by removing RecA filaments from ssDNA (*Lestini and Michel, 2007*). Along with its homologous proteins PcrA and Rep, UvrD has been demonstrated in experiments to translocate on ssDNA progressively 3' to 5' (*Matson, 1986*; *Mechanic et al., 1999*; *Dillingham et al., 2000*; *Myong et al., 2005*; *Fischer et al., 2004*). Structures of UvrD-like SF1 helicase solved so far share a four-subdomain tertiary arrangement (1A/2A/1B/2B) (*Singleton et al., 2007*), including two RecA-like domains (1A/2A) which contain the ATP binding site and are proposed to function as the translocase (*Dillingham et al., 2001*; *Lee and Yang, 2006*), and a flexible domain (2B) which is believed to play a regulatory role in helicase activity (*Lohman et al., 2008*; *Dillingham, 2011*). In particular, the 2B domain is known to adopt different conformations (*Velankar et al., 1999*; *Brendza et al., 2005*; *Jia et al., 2011*; *Nguyen et al., 2017*) and has been proposed to act as a 'molecular switch' controlling UvrD unwinding (*Comstock et al., 2015*).

Combining optical tweezers and single-molecule FRET, *Comstock et al. (2015)* demonstrated that UvrD can switch its activity from DNA unwinding to rezipping (measured by optical tweezers) by dramatically changing its conformation between two states (detected by FRET). The transition from unwinding to rezipping activities was proposed to occur through switching ssDNA strands, accompanied by rotation of the 2B domain (see *Figure 1b*). In this model, the GIG motif on 2B serves as an anchor point on dsDNA above the fork junction, such that rotation of 2B can position the 1A/2A translocase domains on either ssDNA strand, leading to 3' to 5' UvrD translocation either toward (unwinding) or away from (rezipping) the DNA fork. Two crystal structures seem to support this strand-switching model (see *Figure 1c*): one structure of UvrD (pdb code: 2IS2) (*Lee and Yang, 2006*) bound to a dsDNA-ssDNA junction is expected to be the 'unwinding' state (defined here as the 'closed' state) because its 1A/2A domains would translocate UvrD into the DNA fork; the other structure (pdb code: 3LFU) (*Jia et al., 2011*) solved without DNA is expected to represent the 'rezipping' state (defined here as the apo state) because the 1A/2A domains presumably would be bound to the opposing strand, translocating UvrD away from the DNA fork. The structural differences between closed and apo states mainly involve a simple rotation of the 2B domain (*Figure 1c*).

However, in order for the ssDNA strand-switching to happen, the rezipping state must adopt a conformation with a gap between the 1B and 2B domain that is large enough for the bound ssDNA to escape, whereas in both the closed and apo structures the four domains 1B-1A-2A-2B form a closed ring topologically. As we show here, contrary to the common assumption that the apo structure is a functional state of UvrD, the FRET signal simulated using real fluorophores attached to the apo-state structure does not match the experimentally observed signal of the rezipping state, nor the unwinding state. Furthermore, it has been reported that cross-linking the 2B and 1B domains of the SF1 helicase Rep can change it into a superhelicase (*Arslan et al., 2015*), capable of unwinding thousands of base pairs processively. What are the key regulatory factors for the functional switch and is it possible to design mutants with different activities?

To characterize the conformational states of UvrD at the fork junction and the transitions between those states, we use MD simulations, which are well-suited to study atomic-level mechanisms in conjunction with crystallography, single-molecule and biochemical techniques (*Russel et al., 2009*; *Zhao et al., 2010*; *Arkhipov et al., 2013*; *Cheng et al., 2017*; *Latorraca et al., 2017*). However, due to the very long time-scale of conformational changes, brute-force simulations are challenging in the case of large molecular motors such as UvrD. Here, we employed a novel computational approach which integrates advanced sampling simulations with bioinformatics tools that survey

structural information from homologs. We were able to identify modes of motions for function switching from principal component analysis of a 'trajectory' derived from the alignment of various surveyed crystal structures. Using the first two principal components as reaction coordinates, the subsequent all-atom Hamiltonian replica exchange simulations (totaling 12$\mu$s) predict a metastable 'tilted' conformation, which has significantly lower free energy than the apo state. The lowest free-energy path is determined to describe the transition between the closed state to the 'tilted' state. After the closed-to-tilted transition takes place, 2B and 1B domains are separated with enough distance from each other to enable strand-switching to happen. We demonstrate that ssDNA can be disengaged from the ssDNA binding domains of UvrD in the tilted state. Furthermore, the tilted UvrD structure is shown to be able to form stable interactions with the opposing strand after ssDNA strand switching has occurred. We also highlight the role of the GIG motif in assisting 2B domain diffusion along dsDNA during strand-switching. These findings suggest principles underlying mechanisms of related molecular machines beyond what we have known from existing structures.

The properties obtained from the transition pathway are consistent with the single-molecule data (*Comstock et al., 2015*) as well as mutagenesis studies (*Meiners et al., 2014*). Firstly, we carried out equilibrium simulations of UvrD site-specifically labeled with FRET dye pair AlexaF555/AlexaF647 for both the closed state and the tilted state. The calculated average FRET efficiencies for the two states are in good agreement with those for the unwinding and rezipping states measured in single-molecule experiments, respectively. These simulations also allow us to obtain key fluorophore conformations in the tilted state to explain the shape of the experimental FRET distribution. Secondly, we illustrate the molecular basis for hyper-helicase activity of a UvrD double mutant (D403A/D404A) for the first time. Finally, a physical model integrating the simulation results and the measured equilibrium constant from optical tweezers experiments is provided to explain the helicase function-switching mechanism.

## Results

### Structural bioinformatics analysis of conformational ensembles of UvrD-like proteins

Our goal is to characterize UvrD conformational changes that switch its function. Recently, free-energy simulation methods have been successfully applied to study transitions between two functional conformational states of complex molecular machines (*Moradi and Tajkhorshid, 2013*; *Ma and Schulten, 2015*; *Czub et al., 2017*). However, for UvrD all the known structures bound to the DNA fork junction belong to the closed (unwinding) state (*Figure 1c*). It is unclear whether the apo state of UvrD could bind to the dsDNA-ssDNA junction. By aligning the apo state to the closed state, we found geometrical clashes between the fork junction and the apo state (*Figure 1—figure supplement 1b*). We thus forced UvrD at the fork junction to rotate from the closed state to the apo state using targeted molecular dynamics (*Schlitter et al., 1994*) (see *Figure 1—figure supplement 1c* for details). However, such an operation experienced large resistance (DNA was free to move), and the protein returned back to the vicinity of the closed state after the external force was released. We thus need to find new conformations that can represent the rezipping state.

In order to reach the rezipping state while bound to the fork junction, UvrD must reach some hidden metastable states, which can be far away from the 2B-domain-rotation pathway around the dsDNA axis. To identify such states, we developed an approach based on surveying the pdb database (details in Materials and methods). We used protein-protein BLAST (basic local alignment search tool) to search the swissport database with the UvrD sequence as the query sequence. Then, we downloaded the pdb files of these homologs with 40% or more sequence identity. A subsequent principal component analysis (PCA) was carried out to find out the most significant degrees of structural variations among UvrD and its homologs. The coordinates of the homolog structures were then projected onto the first two principal components (PC1 and PC2) (*Figure 2*). Three distinguishable clusters are shown in *Figure 2*: one represents the canonical closed conformation, one represents the canonical apo state, and another one represents an interesting conformation (from the replication initiator protein) in which the 2B domain is tilted from the dsDNA axis. All the structures belonging to the apo state are without nucleic acids bound. The structure in the 'tilted' cluster only has ssDNA bound, and thus very likely it is not a functional state of UvrD because of the absence of

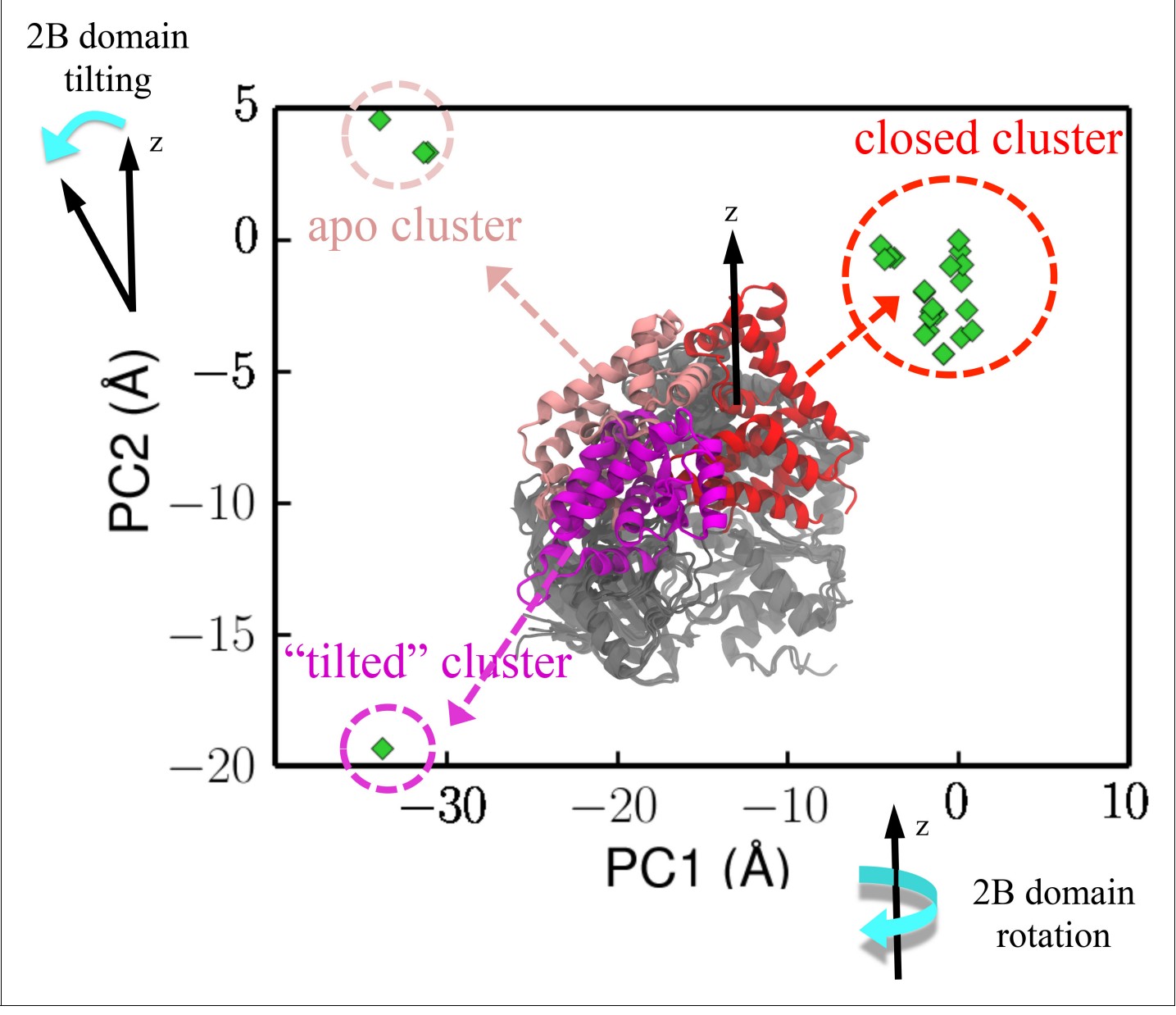

**Figure 2.** Projection of crystal structures onto the first two PCs from PCA. The structures were obtained through a pdb survey. The structures in the middle of the panel show three structural clusters of the 2B domain, labeled red (closed cluster), magenta (tilted cluster) and pink (apo cluster), respectively. The remaining three domains (1A/2A/2B), which are labeled in gray, only have very small structural variation among the homologs. PC1 represents a rotation motion of the 2B domain around the z-axis, whereas PC2 represents a tilting motion away from the z-axis (see *Figure 2—figure supplement 1b*).

DOI: https://doi.org/10.7554/eLife.34186.004

The following figure supplement is available for figure 2:

**Figure supplement 1.** First 2 PCs contribute most of the motion of the 2B domain in the closed to tilted state transition.

DOI: https://doi.org/10.7554/eLife.34186.005

dsDNA interactions. To characterize the functional state for rezipping, we need to carry out all-atom free-energy simulations (the next subsection).

We next calculated the so-called involvement coefficients (*Lei et al., 2009*) (ICs), which are often used to show the contribution of individual modes to the overall structural displacement. For the displacement between the closed structure and the tilted structure, the ICs of the first two PCs are very high (see *Figure 2—figure supplement 1a*), indicating that the first two PCs are sufficient to describe the protein conformational changes based on the available UvrD homolog structures. Directions of motions along the first two PCs are shown on the closed structure (*Figure 2—figure supplement 1b*). We noted that PC1 is in a similar direction as the rotational movement between the closed and apo states. PC2 represents a tilting motion orthogonal to the rotation. Since the closed-to-apo rotation of the 2B domain cannot bring UvrD to the rezipping state due to steric clashes, we suspect that PC2 might make a very important contribution to UvrD conformational switching when bound to the junction.

## Free-energy landscape of UvrD conformational ensembles when bound to the fork junction

Based on the information revealed by the PCA analysis, we would like to find the UvrD conformation responsible for the rezipping state when bound to the dsDNA-ssDNA junction. For this purpose, extensive enhanced sampling simulations (12 $\mu$s in total) were carried out to characterize the free-energy landscape of UvrD conformations and detect any interesting metastable states in it. See Materials and methods for the setup and simulation details.

We first characterized the 2D potential of mean force (PMF) using the first two PCs as coordinates (middle panel of *Figure 3*). We identified two conformations located in the two local minima of the 2D PMF map, respectively (right and left panels of *Figure 3*). These two conformations are defined as 'closed' and 'tilted' states. The tilted state has features that have not been found in any of the existing crystal structures, as we show in the following sections. The PDB file for the newly found tilted state is provided as *Supplementary file 1*.

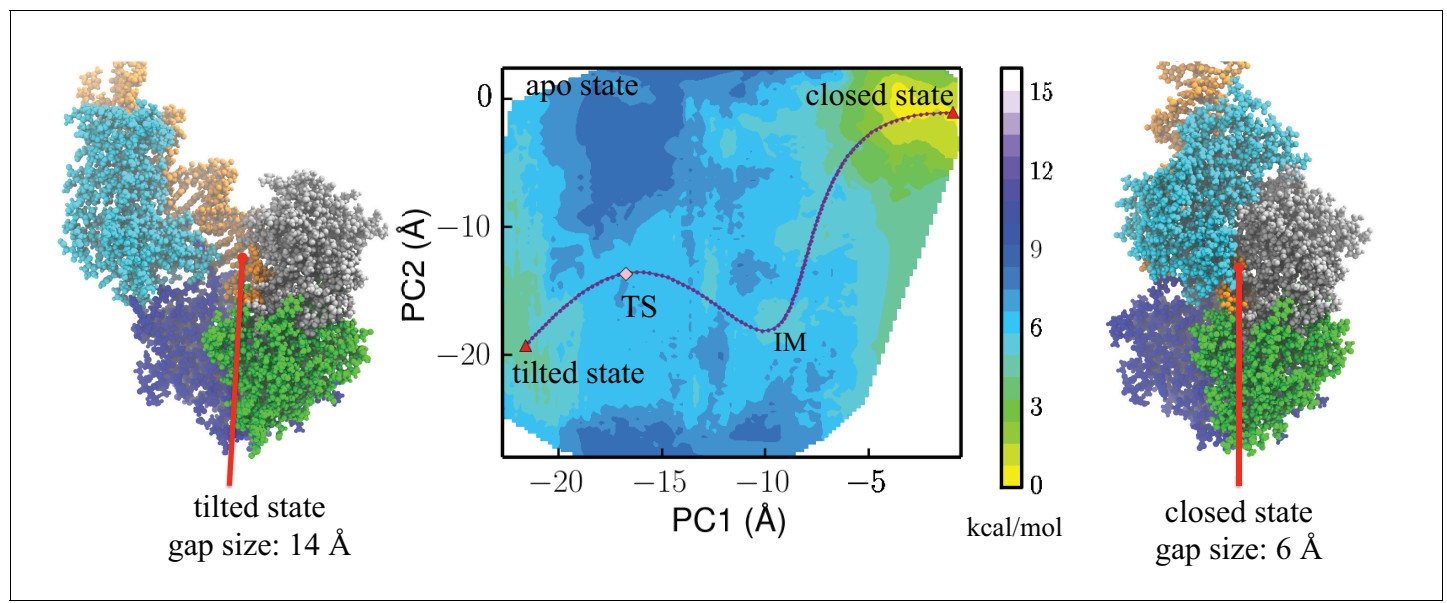

**Figure 3.** Free-energy profile generated using the projections along the first 2 PCs. The transition state (TS) and an intermediate state (IM) are located between the closed-to-tilted transition (the predicted tilted state PDB is provided as *Supplementary file 1*). Right and left panels: snapshots for the closed and tilted states are shown along with the gap size, which is defined by the minimal $C_\alpha$ distance between 2B and 1B domains.

DOI: https://doi.org/10.7554/eLife.34186.006

The following figure supplement is available for figure 3:

**Figure supplement 1.** Free-energy profile and key conformations along the transition pathway.

DOI: https://doi.org/10.7554/eLife.34186.007

The closed and tilted conformations served as the initial and final states for a transition path finding protocol, which was employed to find the lowest free-energy path between them (see Materials and methods). The most probable transition happens in two phases, during which the 2B domain undergoes coupled rotational and tilting motions. In the first phase (closed→IM), 2B carries out a large-scale tilting motion along PC2, overcoming a 4.4 kcal/mol barrier before reaching an intermediate state IM. In the second phase (IM→TS→tilted), 2B performs mostly a rotational motion along PC1, overcoming a 1 kcal/mol barrier ($G_{TS} - G_{IM}$) at the global transition state (TS) before reaching the tilted state. Thus, the rate-limiting step is the first phase, which involves mostly a tilting motion. *Figure 3—figure supplement 1* provides the PMF values and intermediate conformations along the lowest free-energy path. *Video 1* shows the conformational changes of UvrD during the transition.

One can notice that the region the apo structure represents has a high energy value, which is more than 8 kcal/mol higher than the initial state. This demonstrates that the apo state, which is connected to the closed state by 2B domain rotation, is very unfavorable at the dsDNA-ssDNA junction.

We took the representative protein structure in the final and initial states and measured the gap size, which is defined by the closest $C_\alpha$ atom distance between the 2B and 1B domain. The extended ssDNA has a diameter around 10 Å (*Landy et al., 2013*). The initial closed state has a very small gap size of 6 Å, through which the ssDNA cannot pass. The final tilted state has a gap size of 14 Å, which is open enough for ssDNA to pass through.

The overall free-energy landscape projected along a progress variable $\alpha$ is plotted in *Figure 4*. $\alpha$ is proportional to the projection on PC1 and is scaled from 0 to 1.0 between the closed state and the tilted state. The free energy for the metastable tilted state is about 2.5 kcal/mol higher than that of the closed state. The system has to overcome a 4.2 kcal/mol energy barrier at the transition state (TS) to reach the tilted state.

## Validation of the predicted tilted state

We first tested if the ssDNA can escape from the tilted structure. To accelerate the process, we used targeted molecular dynamics by adding a harmonic potential to the coordination number between UvrD and ssDNA. The targeted coordination number was forced to change from an initial value of 18 to 0 in 30 ns. As shown in *Video 2*, the ssDNA is seen disengaged from the ssDNA binding domains of UvrD. The final interaction energy between ssDNA and the 1A/2A/1B domains of UvrD gradually drops to zeros (see *Figure 5—figure supplement 4*). Further below, we also show that this tilted structure can bind stably to the opposing strand to complete the strand-switching process (Figure 8d).

To quantitatively validate our simulation results against experimental data, we compared the FRET efficiency distributions predicted for the closed and tilted states computationally to those of the functional states measured experimentally. We first obtained the 1D FRET efficiency distributions for the unwinding and rezipping state based on the raw single-molecule data (see Materials and methods for details). The distributions, shown in *Figure 5a*, have peak positions at 0.66 and 0.29 for unwinding and rezipping, respectively. By explicitly simulating UvrD in the two states with fluorophore labels (AlexaFluor555/AlexaFluor647) as in the single-molecule experiments, we also determined FRET efficiencies for the closed and tilted states (*Figure 5b*). The simulations accumulated 500 ns for each state, and we considered the orientation factor of the fluorophores in determining the FRET efficiency (Materials and methods).

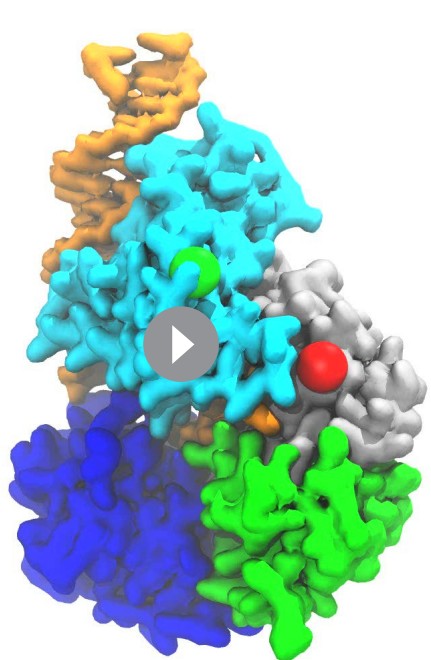

**Video 1.** A movie showing the changes in the molecular structure along the optimal transition path from the closed state to the tilted state.
DOI: https://doi.org/10.7554/eLife.34186.008

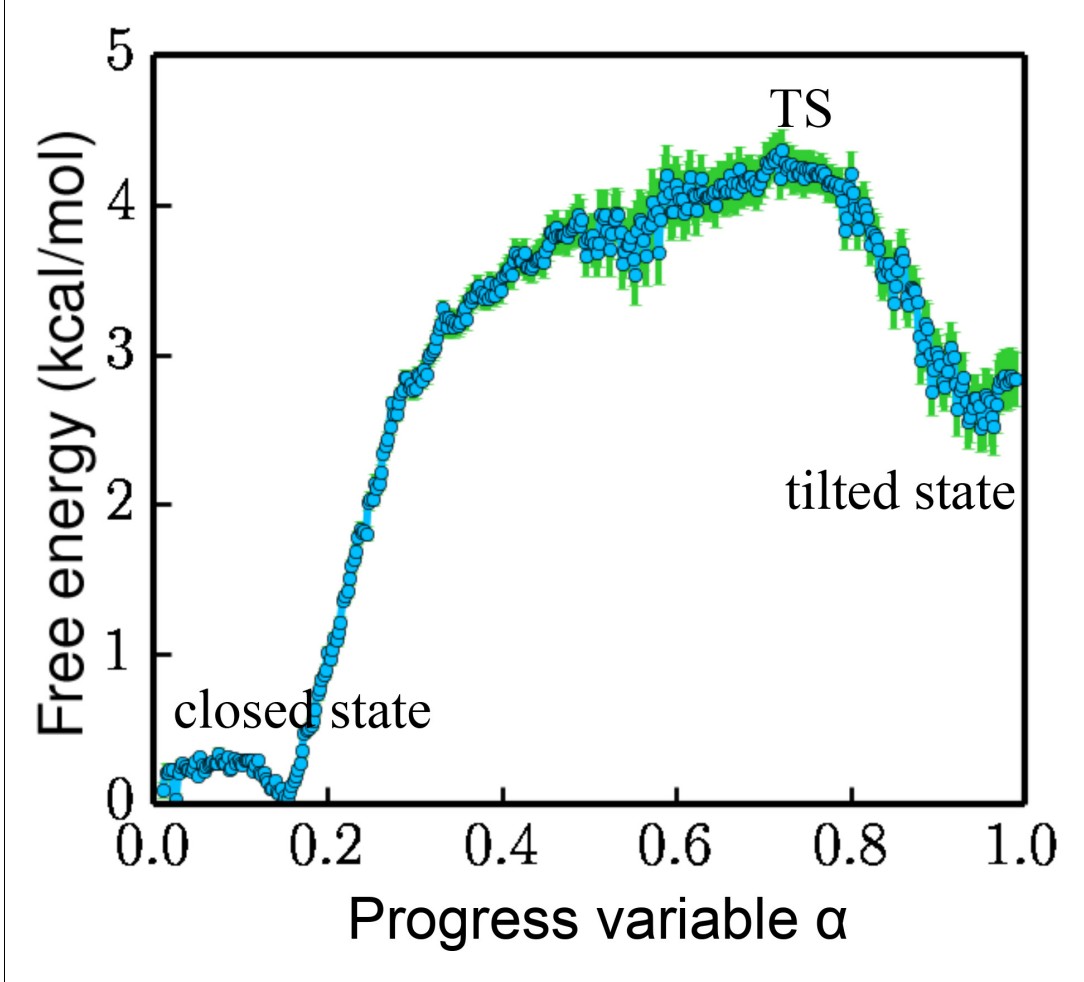

**Figure 4.** Free energy projected along the progress variable $\alpha$. The value of $\alpha$ ranges from 0 to 1.0 (the closed state being 0 and the tilted state being 1.0). The standard error is calculated by a bootstrapping error analysis procedure.
DOI: https://doi.org/10.7554/eLife.34186.009

The predicted FRET efficiency peak for the closed state is at 0.72, whereas the peak for the tilted state is around 0.31. The close agreement between experimental and simulated FRET distributions reaffirms that the tilted state should be the protein conformation responsible for rezipping. As a control, we simulated the apo-state structure with the fluorophore labels for 500 ns as well. The apo-state FRET distribution, which peaks at 0.16, is quite different from the rezipping-state distribution (*Figure 5—figure supplement 1*), suggesting that the apo structure is not the conformation for UvrD rezipping at the junction.

We further examined the representative fluorophore pair conformations at the local maxima (FRET Efficiency = 0.3 and 0.6) of the tilted state FRET distribution (green curve in *Figure 5b*). It appears that the fluorophores have different conformations at the two different FRET values (*Figure 5—figure supplement 3*), due to the conformational dynamics of the dyes with the long linkers. The 'shoulder' of the tilted-state FRET distribution curve at 0.6 efficiency is caused by a metastable conformation of AlexaFluor555 with different pair-distance and orientation comparing to the conformation at 0.3 efficiency.

## UvrD diffusion along dsDNA

In the UvrD functional switching model, the 2B domain of UvrD has to maintain contact with dsDNA; otherwise the protein might disassociate from the fork junction during the ssDNA strand exchanging. It is known that the GIG motif (motif IVc) of UvrD plays a key role in interacting with dsDNA

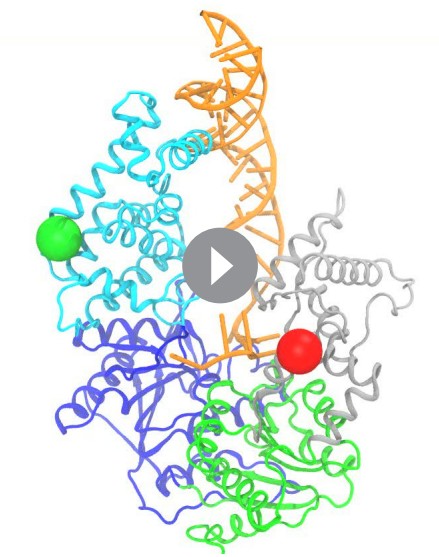

**Video 2.** A movie showing how the ssDNA disengages from its binding domains of UvrD in the tilted state. DOI: https://doi.org/10.7554/eLife.34186.010

(*Myong et al., 2005*), and T422 (a representative residue of GIG) is important for UvrD activity (*Lee and Yang, 2006*). We thus monitored the changes in the interaction between GIG and dsDNA. *Figure 6* shows a free-energy landscape plotted against the DNA base ID in contact with GIG and the distance between them. For each simulation frame, we calculated the distances between every DNA residue's O2P atom and the OG1 atom of T422. Then, the minimal distance and the corresponding DNA base ID were used as the two coordinates. Note the two strands of dsDNA share the same base ID here: for residue x in strand A (indexing according to pdb), the complementary residue in strand B has the same ID x. In the present case, frames with base ID 18 only involve strand A - T422 interaction; whereas frames with base ID 14 only involve strand B - T422 interaction.

In the closed state, residue 18 of strand A contacts the GIG motif, whereas in the tilted state, residue 14 of strand B contacts the GIG motif. Thus, there is a diffusional motion along the dsDNA during the conformational change (see *Figure 6*). In such a way, UvrD is able to switch the binding dsDNA strand and finds an energetically favorable configuration for the ssDNA strand-switching that will happen in the next step. The diffusion happens in a way that the DNA and T422 are disengaged first, and T422 then re-engages with another DNA residue along the double strand. The base ID in contact with T422 during the transition path from the closed state to the tilted state is shown in *Figure 6—figure supplement 1*. One can see that the 2B diffusion happens late during the transition. Although UvrD diffuses along dsDNA during the transition, there is no base pair unwound during the closed-to-tilted transition.

## Molecular mechanism for the UvrD303 mutant

Our simulations provide a molecular explanation for the hyper-activity reported for a mutant (UvrD303) that involves two important aspartic acid residues at the 2B-1B interface. Previous experimental work (*Meiners et al., 2014*) discovered that UvrD303 with substitution of two residues, 403 and 404 (both from Asp to Ala), in the 2B domain exhibits a 'hyper-helicase' unwinding activity in vitro. The authors suggested that such mutations will reduce the 1B-2B domain interactive contacts and thus yield an intermediate conformation instead of a closed conformation. Such an intermediate state they argued would result in the hyperactivity. However, this explanation is not consistent with the single-molecule measurements (*Comstock et al., 2015*) showing that the closed conformation is responsible for unwinding activity.

To reconcile the conflict, we estimated $\Delta\Delta G_{\mathrm{bind}}$ for the binding free-energy between the 1B and 2B domains upon mutating D403 and D404 into alanine, based on our enhanced sampling trajectory. Here $\Delta\Delta G_{\mathrm{bind}} = \Delta G_{\mathrm{bind}}^{\mathrm{mutant}} - \Delta G_{\mathrm{bind}}^{\mathrm{WT}}$, where $\Delta G_{\mathrm{bind}}^{\mathrm{mutant}}$ is the binding free-energy for the mutant and $\Delta G_{\mathrm{bind}}^{\mathrm{WT}}$ is that for the wild type. $\Delta\Delta G_{\mathrm{bind}}$ calculated for the closed state is around $-2.85$ kcal/mol, showing a stabilization effect of the double alanine mutant. On the other hand, $\Delta\Delta G_{\mathrm{bind}}$ calculated for the tilted state is around 0. This indicates that UvrD303 actually favors the closed conformation and thus will lead to better unwinding activity. The so-called MM/PBSA method (molecular mechanics Poisson-Bolzmann surface area) (*Kollman et al., 2000*; *Homeyer and Gohlke, 2012*) was used for calculating $\Delta G_{\mathrm{bind}}$.

*Figure 7a* shows the configuration of D403/D404 and key residues on 1B that contribute most significantly to the binding energy change upon the mutation in the closed state. The first five residues on 1B with the largest contribution to $\Delta\Delta G_{\mathrm{bind}}$ are listed in *Figure 7b* (for the tilted state, all

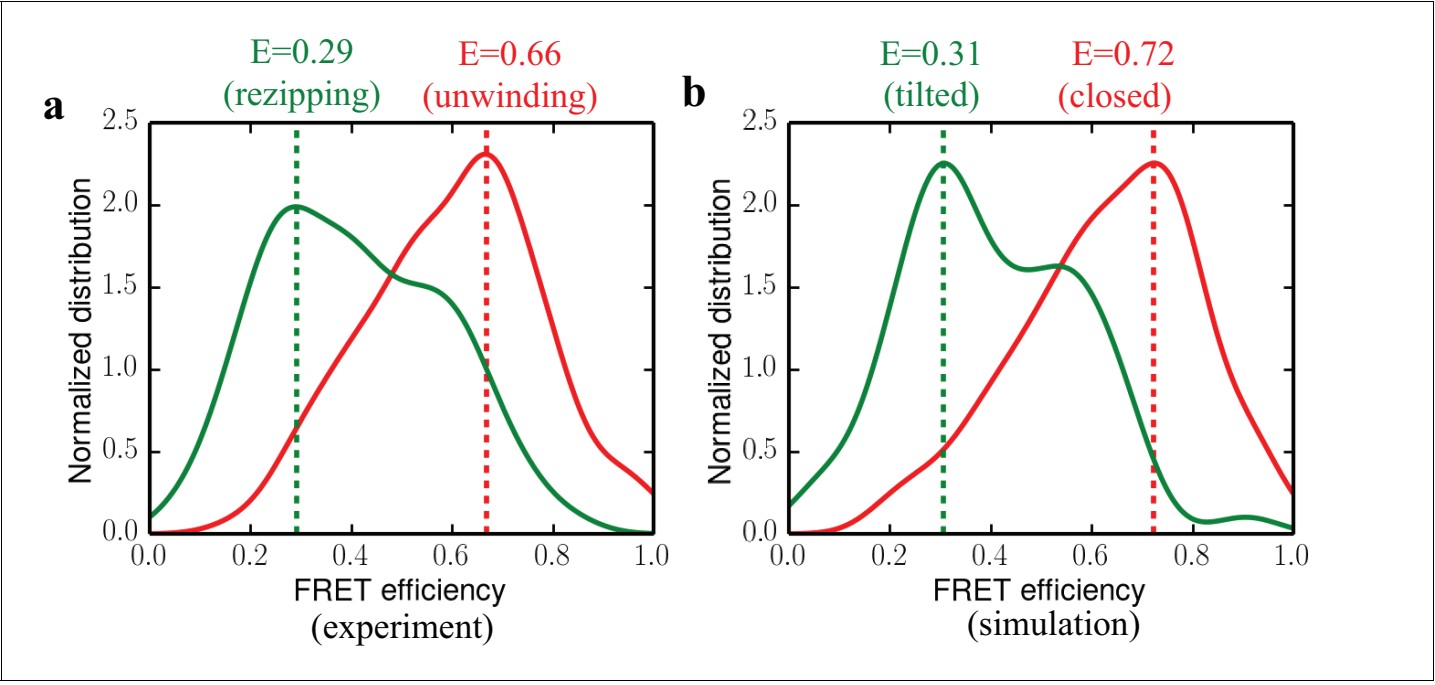

**Figure 5.** Comparing experimental FRET efficiency distributions to the distributions obtained from simulations. (**a**) Experimental distributions for the unwinding and rezipping states. The dotted lines show the peak positions for the two states. (**b**) Simulated FRET efficiency distribution for the closed and tilted states.

DOI: https://doi.org/10.7554/eLife.34186.011

The following figure supplements are available for figure 5:

**Figure supplement 1.** Simulated apo-state FRET distribution (solid cyan curve) and its comparison with the distributions for the rezipping state (dotted green curve, from experiments) and the tilted state (solid green curve, from simulations).

DOI: https://doi.org/10.7554/eLife.34186.012

**Figure supplement 2.** Molecular structures of the dyes (AlexaF555/AlexaF647) used in the simulations.

DOI: https://doi.org/10.7554/eLife.34186.013

**Figure supplement 3.** Averaged fluorophore pair conformations in the tilted state at different FRET efficiency values.

DOI: https://doi.org/10.7554/eLife.34186.014

**Figure supplement 4.** Interaction energy (electrostatic + Van der Waals) between the ssDNA and the ssDNA-binding domains (1A/2A/1B) during the ssDNA disengagement simulation (the protein is restrained in the tilted state).

DOI: https://doi.org/10.7554/eLife.34186.015

the individual residue contributions to $\Delta\Delta G_{\text{bind}}$ become zero). We noted that there are not many positively charged residues on 1B that are very close to D403/D404. The maximum number of hydrogen bonds formed between D403/D404 and the 1B domain is around two pairs during the simulations. Considering that there are also negatively charged residues of 1B (E118/E117) near D403/D404, mutating the two aspartic acid residues into alanine will not decrease but rather increase the interaction strength between 1B and 2B. We also found that there are significant numbers of nonpolar residues located around residues 403 and 404 (L186, A184, L114, I113, L122). Thus, mutating the two charged residues into hydrophobic residues instead increases the interaction strength between the nonpolar groups and the two alanine residues. Overall, the stabilization of the closed state of UvrD303 leads to consistent unwinding of UvrD helicase, reconciling the biochemical measurement (*Meiners et al., 2014*) with the single-molecule experiment (*Comstock et al., 2015*).

## Discussion

We have characterized the conformational dynamics and a key metastable state of UvrD at a fork junction with a hybrid computational approach. The transition pathway as well as the free-energy landscape for UvrD functional switching at the fork junction was obtained, and we found that the

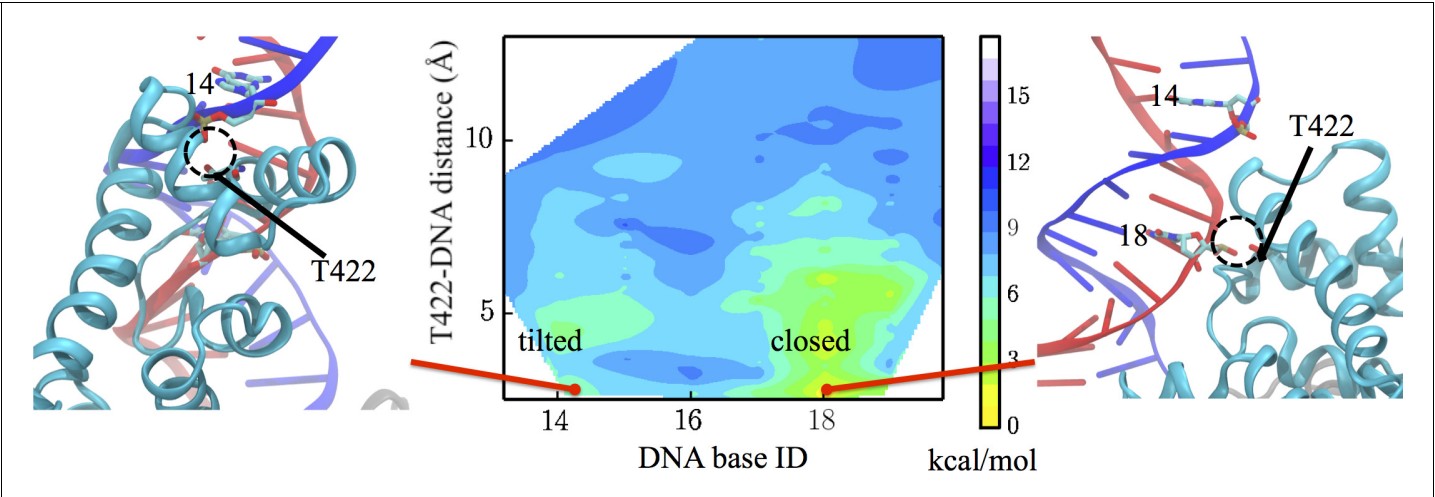

**Figure 6.** Interaction changes between the GIG motif and dsDNA. Here, we use the DNA base ID to represent the closest DNA residue in contact with T422 (part of the GIG motif) on strand A (red) or its complementary residue on strand B (blue). T422 engages with the backbone phosphate of residue 18 of strand A in the closed state, whereas it engages with the phosphate of residue 14 of strand B in the tilted state.
DOI: https://doi.org/10.7554/eLife.34186.016
The following figure supplement is available for figure 6:

**Figure supplement 1.** Evolution of the DNA base ID in contact with the GIG motif during the transition from the closed state to the tilted state (along the lowest free-energy path).
DOI: https://doi.org/10.7554/eLife.34186.017

opening of the 2B domain involves a major tilting motion followed by a major rotational motion. Diffusion of 2B along the dsDNA happens in the late stage of the transition, during which the GIG motif switches its contact from one strand of dsDNA to the other strand. The transition leads to a gap opening between 2B and 1B, which enables the ssDNA to escape presumably allowing the motor domains to strand-switch.

## A physical model for UvrD functional switching

A schematic model can be established based on the simulation results (*Figure 8a,b*). The corresponding molecular models are shown in *Figure 8d*. The UvrD functional switching happens in a two-step manner. A first step is the opening of the 2B domain, followed by a second step of the switching of the bound ssDNA strand, in which the original ssDNA disengages from the 1A/2A/1B domain binding site and the other strand fills in.

To obtain the free-energy difference between the unwinding and rezipping states, we performed a dwell time analysis based on past single-molecule measurements (*Comstock et al., 2015*). The dwell times of the unwinding and rezipping states of UvrD monomers are plotted in a histogram and the calculated averaged rates for both transitions are almost equal ($k_{\text{unwind} \rightarrow \text{rezip}}$ = 6.6 s$^{-1}$ and $k_{\text{rezip} \rightarrow \text{unwind}}$ = 7.0 s$^{-1}$) at 10 $\mu$M ATP (see *Figure 8c*). Thus, the equilibrium constant is around one and the unwinding and the re-zipping conformations should have similar free energy. This is consistent with the picture that the tilted state is a little bit less favorable than the initial state but as soon as the ssDNA releases and the other ssDNA strand binds to the UvrD, the system returns to a lower free energy (the rezipping state) (*Figure 8a*). For the mutant UvrD303, the free energy for the closed state drops around 3 kcal/mol, whereas $G_{\text{tilted}}$ and $G_{\text{rezip}}$ remain the same. The relative stabilization of the closed state leads to more persistent unwinding.

The strand switching is mostly driven by Brownian motion and does not require energy from ATP hydrolysis. To address the possible effect of ATP on strand switching, we (1) analyzed additional data from the optical tweezers experiments and compared the switching rate at two different ATP concentrations and (2) also analyzed the X-ray structures of the closed state with and without ATP. The dwell time distributions at 10 $\mu$M ATP and 2.5 $\mu$M ATP concentration are plotted in *Figure 8c*. The equilibrium constants of switching measured for the two concentrations are very similar (both

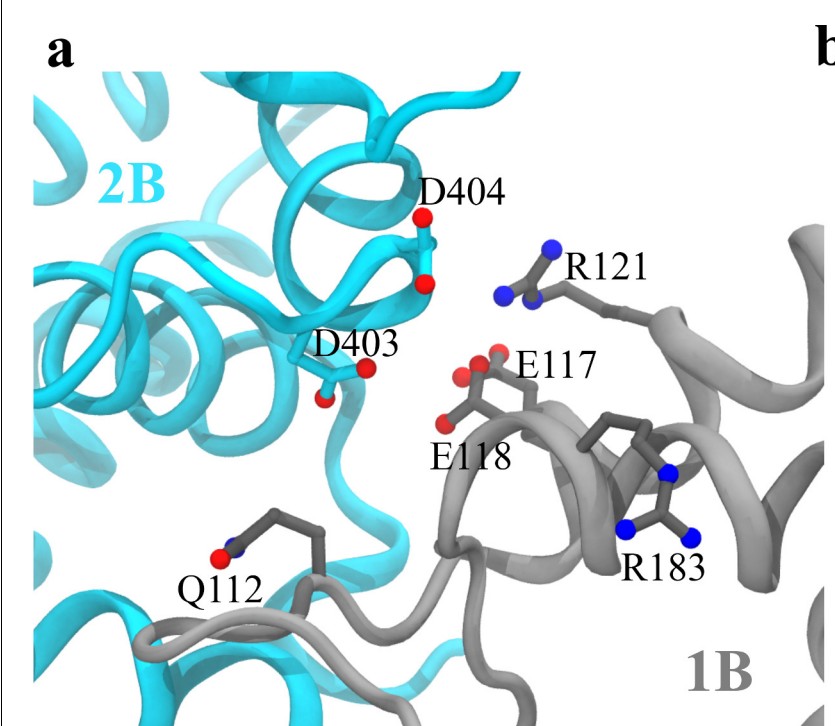

**b** Contribution to $\Delta\Delta G_{bind}$ from different residues of 1B

| Residues of 1B | $\Delta\Delta G_{bind}$ (kcal/mol) |
|---|---|
| All | -2.85 |
| R121 | 0.73 |
| R183 | 0.66 |
| E117 | -0.72 |
| E118 | -1.01 |
| Q112 | -1.38 |

**Figure 7.** Analysis of key interactions at the 1B-2B interface for the UvrD303 mutant. (a) The configuration of key residues involved in the interaction between D403/D404 (belonging to 2B) and the 1B domain. 2B is shown in cyan, whereas 1B is shown in gray. (b) A table showing the contribution to $\Delta\Delta G_{bind}$ from key residues of 1B upon the mutation. Only residues with $|\Delta\Delta G_{bind,x}|>0.6$ kcal/mol are shown, where $x$ is the residue index. Positive values indicate destabilization effects of the mutation; negative values indicate stabilization effects.
DOI: https://doi.org/10.7554/eLife.34186.018

around 1), which suggests that strand switching is likely an ATP-independent process. Furthermore, although our simulated system is based on an ATP-free UvrD crystal structure (2IS2), our computational approach covered the structural information from ATP (or its analogs) bound structures (*Figure 8—figure supplement 1*). One can see that the 2B motion between the ATP-substrate bound and empty UvrD in the closed state is small relative to the large closed-tilted conformational change. Therefore, it is not very likely that ATP binding has a noticeable impact on the closed-to-tilted transition.

## DNA-UvrD conformation at the rezipping state

To explore the structure of the rezipping state further, we built a rezipping structure starting from the tilted conformation after ssDNA strand switching has occurred (see *Figure 8d* and Materials and methods). After a 100 ns equilibration simulation, the modeled system was stable and had a rmsd around 3 Å from the tilted state (*Figure 8—figure supplement 2*). The newly obtained rezipping structure satisfies the following considerations: (1) The protein conformation is very similar to the tilted conformation. (2) The interaction configuration between 2B and dsDNA remains the same between the tilted state and the rezipping state. Note that during our simulations of the closed to tilted transition, the 2B domain changed its contact from one strand of dsDNA to the other (4 bp shift, from the red strand A to the blue strand B in *Figure 6*). (3) The ATPase domains 1A-2A are in the correct orientation along the ssDNA (3' to 5'), pointing away from the junction. Such a conformation enables UvrD to translocate along ssDNA, allowing the duplex to rezip behind it. In examining the rezipping structure, we found a small loop forming between the dsDNA junction and the ssDNA-1A binding site. A similar feature was proposed by a translocation model of PcrA helicase in *Park et al. (2010)*, which suggested that PcrA can extrude a ssDNA loop while it attaches to dsDNA and translocates the 5' ssDNA tail in an open conformation.

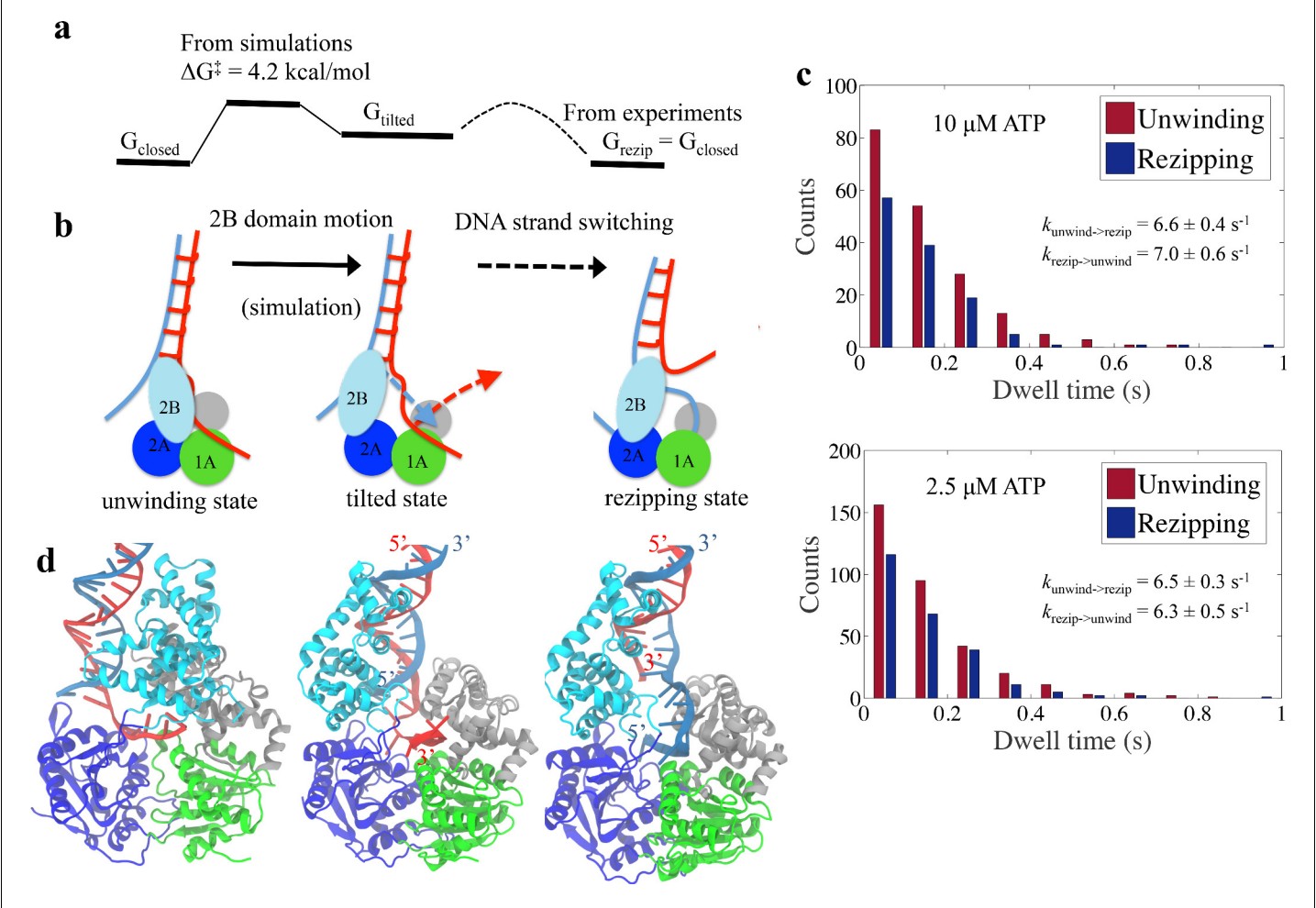

**Figure 8.** Physical mechanism of UvrD functional switching. (a) Illustration for the whole free-energy landscape of ssDNA strand switching enabled by UvrD conformational transition. (b) Schematic representation showing the 2-step process of how UvrD switches the ssDNA strand along which the motor domain walks. (c) Dwell time distributions for the unwinding and rezipping states at 10 $\mu$M and 2.5 $\mu$M ATP concentration based on the measured traces from optical tweezers (Materials and methods). (d) The structural models for the unwinding, tilted and rezipping states are shown from left to right. The structural model for the rezipping state was obtained from the tilted state after the ssDNA strand switching as illustrated in Methods. Strands A and B of the dsDNA are shown in red and blue, respectively.

DOI: https://doi.org/10.7554/eLife.34186.019

The following figure supplements are available for figure 8:

**Figure supplement 1.** Projection of crystal structures onto the first two principal components from PCA (same data points used in *Figure 2*).

DOI: https://doi.org/10.7554/eLife.34186.020

**Figure supplement 2.** Rezipping-state RMSD (calculated using the protein non-hydrogen atoms) from the tilted state during the 100 ns equilibration simulation.

DOI: https://doi.org/10.7554/eLife.34186.021

**Figure supplement 3.** The apo structure is not accessible after the closed-to-tilted transition at the DNA junction.

DOI: https://doi.org/10.7554/eLife.34186.022

The apo state seen in the crystal structures without DNA bound is likely not a functional state of UvrD at the fork junction. First, the simulated apo-structure FRET distribution is quite different from the rezipping-state FRET distribution. Upon completion of the ssDNA strand-switch, we expect the conformation of UvrD to stay close to the tilted structure. The FRET signal from the single-molecule experiment shows a clear two-state distribution, and our FRET distribution for the tilted state from simulations agrees very well with the experiment. Second, we aligned the apo state structure to the tilted state in *Figure 8—figure supplement 3*, and there are serious clashes between the apo

structure and the dsDNA. Thirdly, the apo state is highly unfavorable at the fork junction according to our simulations.

## Functional insights for UvrD and its homologs

Our simulations, backed by the single-molecule measurements, provide functional insights for UvrD in several important biological processes. For example, frequent strand switching of UvrD due to 2B conformational transition results in unwinding over short distances (*Comstock et al., 2015*), which is consistent with the small number of basepairs unwound during nucleotide excision repair (*Kisker et al., 2013*). On the other hand, UvrD303 is associated with a recombination-deficient phenotype (*Centore et al., 2009*), possibly due to lacking such a structural transition as the closed state is over-stabilized. It has been reported that UvrD can dismantle RecA filaments from the ssDNA at a stalled replication fork (*Veaute et al., 2005*; *Lestini and Michel, 2007*). As RecA has a central role in homologous recombination (*Cox, 2007*), a population shift toward the closed state could enhance UvrD's ability to disrupting RecA-ssDNA filaments and impair recombinational repair.

The tilted state and related motions found here can possibly help connect structural information with function for other SF1 helicases. A highly homologous helicase, PcrA, is known to efficiently strip RecA filaments off ssDNA in an 'open'conformation (*Park et al., 2010*). The low-FRET 'open' conformation of PcrA could be similar to the tilted conformation revealed in this study. In this case, PcrA is anchored to the dsDNA and translocates the 5' ssDNA strand in the direction toward the junction. A different mode of PcrA is binding to the 3' ssDNA and the dsDNA while unwinding the duplex in the closed form (*Velankar et al., 1999*; *Niedziela-Majka et al., 2007*). Another UvrD homolog RecB, by mostly tilting its 2B domain from the putative closed state, forms interactions with other subunits in the RecBCD complex (*Singleton et al., 2004*; *Wilkinson et al., 2016*), which has a key role in initiating recombinational repair (*Spies and Kowalczykowski, 2005*).

It may be possible to engineer UvrD-like helicases with tunable unwinding activities. Experiments have shown that cross-linking Rep and PcrA in the closed form resulted in superhelicase activity (*Arslan et al., 2015*). We demonstrated that mutating the two aspartic acid residues into alanine on 2B domain stabilizes the UvrD closed conformation. The analysis of the binding free-energy change upon the mutation (*Figure 7b*) provides potential target residues to guide future experimental designs. For example, mutating some negatively charged residues on 1B might also result in hyper-helicase behavior. Our findings for the conformational dynamics of UvrD and the related computational strategy establish a foundation for future studies to reveal principles employed by other related helicase systems.

# Materials and methods

## Structural bioinformatics analysis of UvrD homologs

Our computational study starts from analyzing the structural ensemble of UvrD homologs. There are two representative conformations for UvrD: one being the so-called 'closed' state (e.g. 2IS2); the other one being the apo state (e.g. 3LFU). As stated in Results, the apo state is likely not a functional structure of UvrD at the DNA fork junction. To explore the conformational space of UvrD as much as possible, we performed a structural survey for possible UvrD homologue conformations using bioinformatics sequence and structure alignment tools (*Altschul et al., 1997*; *Cock et al., 2009*; *Bakan and Bahar, 2009*; *Bakan et al., 2014*). The initial sequence alignments were obtained using NCBI blastp search (*Altschul et al., 1997*) of Protein Data Bank database sequences, with UvrD as the query sequence. Twenty-six structures were selected from the surveyed structures with sequence identity better than 40% and query sequence coverage larger than 60%. The structure alignment was generated by ProDy (*Bakan and Bahar, 2009*; *Bakan et al., 2014*) from the pairwise sequence alignments by Biopython (*Cock et al., 2009*). The resulting 26 structures can be interpreted in terms of a 'trajectory' with the coordinates $\mathbf{r}(k) = (r_1(k), r_2(k), ..., r_{3N}(k))^\top$, of which each frame $k$ ($k = 1, 2, ..., 26$) contains $3N$ coordinates (from $N$ $C_\alpha$ atoms) of the homologous structures that were mapped onto the original UvrD chain.

We then performed principal component analysis (PCA) (*García, 1992*; *Bakan and Bahar, 2009*; *Raveh et al., 2016*) with ProDy to determine a number of modes for reducing the phase space of UvrD motion. Only the $C_\alpha$ coordinates of the 2B domain were used for the PCA calculations, after

aligning the 1A/2A/1B domains of all the 26 structures to those of the closed state structure (2IS2). The covariance matrix $\sigma$ for PCA is determined via $\sigma = \langle (\mathbf{r}(k) - \langle \mathbf{r}(k) \rangle)(\mathbf{r}(k) - \langle \mathbf{r}(k) \rangle)^T \rangle$, where the angular brackets $\langle \rangle$ denote the average over $k$ (all the frames). The eigenvectors $\mathbf{v_i}$ (principal components or PCs) of the $\sigma$ matrix are determined by $\lambda_i \mathbf{v_i} = \sigma \mathbf{v_i}$. These PCs, which are ranked by their corresponding eigenvalues, represent different directions of conformational motion away from the original closed state.

The homologous structures were then projected onto the first two PCs with the largest eigenvalues. As stated in Results, a 'tilted' structure based on pdb 1UAA was found as an outstanding cluster among the homologous structures. To see the contributions of different PCs to the displacement between the closed structure and tilted structure, we further calculated the involvement coefficient $\eta_i$ (*Ma and Karplus, 1997*; *Lei et al., 2009*) of the $i$th PC. $\eta_i$ is defined as $|\mathbf{v_i} \cdot \Delta\mathbf{R}|$, where $\Delta\mathbf{R}$ is the unit vector describing the displacement from the closed structure to the tilted structure. Only the first two PCs contribute significantly to the overall motion (*Figure 2—figure supplement 1A*). PC1 and PC2 are used later as coordinates to compute the free-energy landscape.

## MD simulation setup

Our simulations were initiated from the closed state (pdb 2IS2) of UvrD (see *Figure 1c*). The protein-DNA system was solvated in a 100 Å ×100 Å ×130 Å water box with 55 mM NaCl (the system had ∼140K atoms in total). A $2 \times 10^4$-step energy minimization was carried out and the system was then heated to 310 K in 30 ps, employing harmonic constraints with 1 kcal/(mol Å²) spring constant to the $C_\alpha$ atoms. Keeping the spring constant, a 1 ns equilibration in the NPT ensemble (1 atm at 310 K) was performed with a Langevin thermostat for temperature coupling. This was followed by a 1 ns NVT-ensemble simulation, during which the spring constant was gradually decreased to zero. The system was then equilibrated for 60 ns, and the resulting configuration is referred to as the closed state. All MD simulations in our study were performed using NAMD 2.10 (*Phillips et al., 2005*) with the CHARMM36 force field (*Best et al., 2012*; *Hart et al., 2012*).

## Free-energy simulation protocol

To determine the free-energy profile along a reaction coordinate, we employed the Hamiltonian replica-exchange (HREX) method (*Park et al., 2012*; *Jiang et al., 2012*; *Jiang et al., 2014*). HREX uses a series of replicas ($j = 1, 2, .., $ M) of the system, which are simulated concurrently with slightly different Hamiltonians and are exchanged frequently among themselves based on the Metropolis exchange criterion (*Sugita et al., 2000*). HREX can be very powerful in reconstructing rugged free-energy landscapes by exchanging external biasing potentials, which, with different biasing parameters, are added to the replicas to enhance the sampling throughout the reaction coordinate (RC). The biasing potential (or the window potential) for each replica $j$ usually assumes the form of $U_m(\xi_j) = k_m(\xi_j - p_m)^2/2$, where $\xi_j$ is the current value of the reaction coordinate for replica $j$, $m$ ($m = 1, 2, .., $ M) is the index for the biasing potentials (windows), $p_m$ is the preassigned parameter for the center of the harmonic potential, and $k_m$ is the spring constant. The centers of the biasing potentials ($p_m$) are selected as an ordered list of values ($p_1 < p_2 < ... < p_M$) all over the RC to cover the reaction of interest fully. Exchanges between two neighboring replicas (replicas with neighboring $p_m$ values) are attempted periodically during the simulations. Without the replica-exchange strategy, this protocol reduces to the conventional umbrella sampling, which often suffers from the inefficient sampling of degrees of freedom orthogonal to the reaction coordinate (*Jiang et al., 2012*).

The present study chooses the projection on the first PC ($\mathbf{v_1}$) as the reaction coordinate $\xi$ and includes M = 120 biasing windows between the closed state and the tilted state. The initial configurations for the M windows were generated through a 5 ns targeted MD simulation (*Schlitter et al., 1994*), by driving UvrD from the closed state to the tilted state. An exchange between two neighboring replicas was attempted every 10 ps and the spring constant of the harmonic potential was set to 100 kcal/(mol Å²). The production run of each replica lasted 100 ns, and the total simulation time added up to 12 $\mu$s (100 ns × 120). Eventually the weighted histogram analysis method (WHAM) (*Kumar et al., 1992*) was applied to obtain the unbiased 1D and 2D free-energy landscapes in *Figures 4* and *3*. We performed the Monte Carlo bootstrap error analysis (*Stine, 1989*; *Hub et al., 2010*) to estimate the uncertainty along the reaction coordinate. The basic idea of bootstrapping is to obtain several estimates (we obtained 10 trials) for the free energy based on randomly generated

subpopulations from the histogram in each window. Our simulations with HREX benefitted from a scalable multiple copy algorithm (*Jiang et al., 2014*) which enables simulating hundreds of replicas simultaneously on a petascale supercomputer.

As stated in Results, the tilted state structure was identified as one of the most important metastable states. Based on the free-energy landscape using the projections on the first two PCs, the lowest free-energy path describing the most probable reaction mechanism was localized between closed state and the tilted state using the optimization algorithm in (*Ensing et al., 2005*). The path was then smoothed and 120 images were chosen uniformly along the 2D pathway applying the curve-fitting protocol in (*Ma and Schulten, 2015*).

## FRET efficiency calculation based on simulations with dye molecules

To check if the simulated closed and tilted states generate the FRET signals of the respective unwinding and rezipping states measured by the single-molecule experiments, we carried out equilibrium simulations with the actual dye molecules for both states. AlexaFluor555 and AlexaFluor647 maleimides (Molecular Probes, Eugene, OR) were modeled according to (*Vrljic et al., 2010*) and (*Gust et al., 2014*) (see *Figure 5—figure supplement 2*). Then the two dyes were, respectively, attached to UvrD residues 473 and 100, which were mutated to cysteine from alanine. Force field parameters for the dyes linked to a cysteine residue were obtained from the CHARMM General Force Field (CGenFF) (*Vanommeslaeghe et al., 2010*) using the ParamChem server. The total charges were set to 0 and $-3$ for the two dyes respectively (*Gust et al., 2014*). Partial charges on the atoms were further refined by the Force Field Toolkit (ffTK) (*Mayne et al., 2013*) in VMD (*Humphrey et al., 1996*). Parameters for bonds, angles and dihedrals from CGenFF with high penalty scores were validated or refined by ffTK.

To sample dye dynamics efficiently, we launched 50 independent standard MD simulations with random initial velocity seeds for the closed, tilted and apo states. Every single simulation lasted 10 ns and a total 500 ns simulation time was accumulated for each state.

The FRET efficiency was determined by $E = R_0^6/(R^6 + R_0^6)$, where $R$ is the distance between the donor and acceptor, and $R_0$ is the Föster radius (or the 50% energy transfer distance). $R_0$ is given by the relationship (*Wu and Brand, 1994*) $R_0 = \left(8.79 \times 10^{-5} n^{-4} \phi_D J \kappa^2\right)^{1/6}$, where $n$ is the index of refraction, $\phi_D$ is the donor quantum yield, $J$ is the spectral overlap integral, and $\kappa^2$ is the orientation factor. $R_0$ is determined to be 51 Å when $\kappa^2$ equals 2/3, assuming that the dyes randomize their orientations by rapid diffusion prior to energy transfer. Such an assumption can be problematic, and in the present study the orientation factor is calculated using $\kappa^2 = \left(\cos \theta_T - 3 \cos \theta_D \cos \theta_A\right)^2$, where $\theta_T$ is the angle between the donor and acceptor transition dipole moments and $\theta_D$ and $\theta_A$ are the angles between these two dipoles and the vector connecting the donor and acceptor (*Corry and Jayatilaka, 2008*). The transition dipole moments for AlexaFluor555/647 or very similar dyes have been determined in (*Corry and Jayatilaka, 2008*) and (*Graen, 2009*). The simulated FRET data were integrated to four ns per point to obtain its probability distribution using the density kernel estimation method (*Parzen, 1962*; *Comstock et al., 2015*).

## Analysis of single-molecule data

To validate our simulation results, analysis based on the raw data from single-molecule experiments was carried out. *Comstock et al. (2015)* combined optical tweezers (to detect UvrD unwinding activity) and single-molecule FRET (to detect UvrD conformation) measuring both simultaneously. Example raw time traces of UvrD activity and conformation are shown in Fig. 3 and Fig. S5 in *Comstock et al. (2015)* (at 10 $\mu$M ATP concentration). Time traces from the optical tweezers were sampled at 267 Hz. Time traces for donor and acceptor intensities were integrated to 30–60 ms per data point. The time-dependent FRET efficiency $E(t)$ was calculated by $E(t) = 1/(1 + \gamma(I_D(t)/I_A(t)))$ (*Ha et al., 1999*; *Choi et al., 2010*), where $I_D(t)$ and $I_A(t)$ are the measured donor and acceptor intensities, and $\gamma$ is a correction factor accounting for the different detection efficiencies for the two dyes, and can be measured from photobleaching events. $\gamma = \Delta I_A/\Delta I_D$ is determined to be 0.78 from 20 acceptor photobleaching events, where $\Delta I_A$ and $\Delta I_D$ are the acceptor and donorintensity changes upon acceptor photobleaching, respectively.

To measure the FRET efficiency distribution for the unwinding and rezipping states individually, we needed to assign each raw data point to the two states separately. Since the helicase velocity

and FRET efficiency were measured concurrently, we used helicase velocity to define whether each data point in the traces belonged to unwinding or rezipping states (see Fig. S5 in [*Comstock et al., 2015*]). Time intervals were determined during which the helicase was either unwinding, rezipping, or paused (positive velocity indicates UvrD is in the unwinding state; negative velocity indicates rezipping; absolute unwinding velocity smaller than 20 bp/s indicates a pause). Paused states were not considered in the analysis. FRET efficiencies over each time interval were collected for the unwinding and rezipping states from 141 time intervals (13 molecules in total). We then used the density kernel estimation method to obtain the experimental FRET distribution (*Parzen, 1962*; *Comstock et al., 2015*). A density kernel plot is a summation of small Gaussians centered at each FRET data point. We used a standard deviation of 0.06 for the Gaussians.

We also analyzed the dwell times for both the unwinding state (high FRET) and the rezipping state (low FRET) of UvrD monomers. For this purpose, the duration of each time interval defined above was measured using the traces from optical tweezers measurements. We chose to select intervals and calculate the dwell time using the tweezers signal because it has a higher time resolution than the FRET signal (about one order of magnitude higher). The dwell time distribution was obtained by histogramming the collected duration values for the unwinding and rezipping state separately. In order to assess the effect of ATP concentration on UvrD functional switching, we analyzed optical tweezers data of UvrD activity at two different ATP concentrations. *Figure 8c* plots the distributions of dwell times at 10 $\mu$M and 2.5 $\mu$M ATP concentration. The rates of the transitions were estimated by the inverse of the averaged dwell times.

## Modeling of the rezipping-state structure

To construct a structure of the rezipping state (after ssDNA strand switching) starting from the tilted conformation (*Figure 8d*), we consider the following constraints: (1) the 2B domain maintains contact with the dsDNA while the ssDNA binding domains (1A and 2A) disassociate from one ssDNA strand and bind to the other ssDNA strand. Otherwise, the entire protein would dissociate from DNA. (2) The interaction configuration between ssDNA and the motor domains (1A-2A) must remain the same after strand switching. The motor domains move from 3' to 5' on the ssDNA in both the unwinding and rezipping modes. With these considerations, we created a structural model in which we repositioned the UvrD-bound ssDNA segment from the 3' ssDNA tail (strand A) to the 5' ssDNA tail (strand B) of the junction in the tilted state. This was achieved by attaching the 5' terminus of strand B to the 3' terminus of strand A, and by cutting the ssDNA (strand A) at the junction position. We then equilibrated the modeled system in a water box with 55 mM NaCl for 100 ns.

## Acknowledgements

The present contribution is dedicated to Klaus Schulten (1947–2016), whose visionary developments in high performance simulation tools for molecular dynamics permit characterizing biomolecular motor action at biologically relevant timescales. This work has been supported by grants from the NIH (9P41GM104601 to ZL-S and KS; R01 GM120353 to YRC) and from the NSF PHY-1430124 (Center for the Physics of Living Cells to YRC, ZL-S and KS). The authors gladly acknowledge supercomputer time provided by the Texas Advanced Computing Center via Extreme Science and Engineering Discovery Environment grant NSF-MCA93S028 and the Blue Waters sustained-petascale computing project, which is supported by NSF (OCI-0725070 and ACI-1238993) and the State of Illinois. We also thank former Chemla lab member Dr. Matthew J Comstock for scientific discussions.

## Additional information

### Funding

| Funder | Grant reference number | Author |
| --- | --- | --- |
| National Institute of General Medical Sciences | 9P41GM104601 | Zaida Luthey-Schulten Klaus Schulten |
| National Science Foundation | PHY-1430124 | Yann R Chemla Zaida Luthey-Schulten Klaus Schulten |

| National Institute of General Medical Sciences | R01 GM120353 | Yann R Chemla |

The funders had no role in study design, data collection and interpretation, or the decision to submit the work for publication.

## Author contributions

Wen Ma, Conceptualization, Data curation, Formal analysis, Investigation, Methodology, Writing—original draft, Writing—review and editing; Kevin D Whitley, Formal analysis, Writing—review and editing; Yann R Chemla, Conceptualization, Supervision, Funding acquisition, Methodology, Writing—review and editing; Zaida Luthey-Schulten, Conceptualization, Data curation, Supervision, Funding acquisition, Writing—review and editing; Klaus Schulten, Conceptualization, Funding acquisition, Methodology

## Author ORCIDs

Wen Ma http://orcid.org/0000-0002-1123-5273
Yann R Chemla http://orcid.org/0000-0001-9167-0234
Zaida Luthey-Schulten http://orcid.org/0000-0001-9749-8367

## Decision letter and Author response

Decision letter https://doi.org/10.7554/eLife.34186.030
Author response https://doi.org/10.7554/eLife.34186.031

# Additional files

## Supplementary files

• Supplementary file 1. The PDB file for the predicted tilted-state structure of the UvrD-DNA complex.
DOI: https://doi.org/10.7554/eLife.34186.023

• Supplementary file 2. CHARMM36 force field parameters for the fluorophore molecules used in the simulations (AlexaFluor555 and AlexaFluor647).
DOI: https://doi.org/10.7554/eLife.34186.024

• Transparent reporting form
DOI: https://doi.org/10.7554/eLife.34186.025

## Data availability

The PDB file of our predicted structure (tilted state) has been uploaded as Supplementary File 1. CHARMM36 force field parameters for the fluorophore molecules used in the simulations have been uploaded as Supplementary file 2.

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
