## [Decision Letter]

Thank you for submitting your article "Free energy simulations reveal molecular mechanism for functional switch of a DNA helicase" for consideration by *eLife*. Your article has been favorably evaluated by John Kuriyan (Senior Editor) and three reviewers, one of whom is a member of our Board of Reviewing Editors. The following individuals involved in review of your submission have agreed to reveal their identity: Maria Spies (Reviewer #2) and Wei Yang (Reviewer #3).

The reviewers have discussed the reviews with one another and the Reviewing Editor has drafted this decision to help you prepare a revised submission.

Summary:

In this work the authors investigate the conformational change and the underlying energetics of UvrD, a prototypical SF1 DNA helicase, in switching translocating DNA strands. The authors combine molecular dynamics simulation, structural bioinformatics, and re-analysis of experimental data. The manuscript revealed a previously unknown metastable "tilted" UvrD conformation and showed how the so-called 2B domain embedded in the helicase motor core may assert direction control in the switch.

Essential revisions:

While the reviewers agree that this work merits publication in *eLife*, they raised several important issues that need to be addressed:

1) To help the readers who are not familiar with DNA helicases, a general discussion of its function from a structural perspective should be added in the Introduction. A cartoon figure would be helpful.

2) The authors used an Hamiltonian replica exchange method for the free energy calculation. The reliability of the calculation requires at least one structural replica to hop among essential window states with a "round-trip" manner. It is suggested that the authors demonstrate that in at least one single "structural trajectory", the important basins characterized on the free energy surface have been repeatedly visited. This would make the calculations more technically sound.

3) More details should be presented in the main text concerning the retrospective analysis of the FRET states, since this analysis is not commonly performed and it is one of the interesting aspects of this work.

4) A strong agreement between the computed FRET distributions for the "tilted" state and the experimental FRET distributions for the re-zipping state are consistent with the authors' proposal that the UvrD conformation during re-zipping resembles the "tilted" state. In a different part of the text, the authors also compare the free energies of the unwinding and re-zipping conformations, which they expect to be similar because UvrD spends equal amount of time in the two conformations. Since the energy of the tilted state is higher than the closed state (the authors even refer to it as a metastable state), it is unlikely to represent the re-zipping state. In fact, the authors introduce a distinct re-zipping state in the Discussion and in Figure 8A, but its structure is not discussed. Please reconcile the two arguments.

5) The apo state of UvrD was ruled out as a re-zipping configuration due to the clashes with the DNA duplex. Please discuss whether these clashes can be removed by allowing the duplex part of the DNA substrate to move/tilt? Transition from the closed conformation into the tilted state results in a substantial movement of the duplex (obvious from Video 1). Perhaps, further movement of the duplex, or positioning of the motor domains on the 5'-terminated strand (which would be consistent with the re-zipping direction) may allow for the DNA bound apo-like state? Modeling such a structure may be beyond the scope of this paper, but some discussion may be informative.

6) All the structures considered here were ATP free. Please state whether there is a reason to believe that the strand switching takes place in the nucleotide-free state?

---

## [Author Response]

Essential revisions:While the reviewers agree that this work merits publication in eLife, they raised several important issues that need to be addressed:1) To help the readers who are not familiar with DNA helicases, a general discussion of its function from a structural perspective should be added in the Introduction. A cartoon figure would be helpful.

We are happy to provide a general discussion on helicase function from a structural perspective. The following text has been added to the Introduction: “DNA helicases can unwind double-stranded DNA (dsDNA) into single-stranded DNA (ssDNA), which are later copied during DNA replication or modified in DNA repair processes (Wu and Spies, 2013; Lohman et al., 2008). […] Such translocation happens in a stepwise manner, during which the chemical energy from ATP hydrolysis is used to break the bonds in dsDNA via conformational changes of the motor domains (Yang, 2010; Patel and Donmez, 2006).” We also added a new panel to Figure 1 in the main text.

2) The authors used an Hamiltonian replica exchange method for the free energy calculation. The reliability of the calculation requires at least one structural replica to hop among essential window states with a "round-trip" manner. It is suggested that the authors demonstrate that in at least one single "structural trajectory", the important basins characterized on the free energy surface have been repeatedly visited. This would make the calculations more technically sound.

To better discuss the reliability of the Hamiltonian replica exchange method in our study, we need to first state that Hamiltonian replica exchange has been developed to improve the sampling efficiency of the traditional umbrella sampling methods (Jiang, et al., 2012). The convergence of the umbrella sampling simulations can be checked by inspecting the sampling overlap between neighboring umbrella windows (which we provide below) and by estimating the uncertainty. We presented the uncertainty along the transition pathway in Figure 4 in the main manuscript. If there were poor overlaps between neighboring windows the uncertainty would be large.

To show the overlap between the different umbrella windows, in Author response image 1, we plot conformation distributions along the reaction coordinate sampled in windows near the transition state (TS) in Figure 4 of the main manuscript. Because TS has the highest free energy, this region requires the most extensive sampling and has to be checked for the sampling overlap. In the Hamiltonian replica exchange, the umbrella biasing potentials are being exchanged. A histogram with respect to a specific umbrella window was obtained by extracting the frames (from all the replica trajectories) experiencing this specific umbrella window potential. As we can see in the plot, the histograms from the neighboring windows overlap well and can be used to reconstruct the free energy landscape.

The uncertainty measure shown in Figure 4 of the main text was obtained by applying the Monte Carlo bootstrap error analysis (Stine, An Introduction to Bootstrap Methods, 1989; Hub, et al., 2010) and is described in the Materials and methods. The basic idea of bootstrapping is to obtain several estimates for the free energy based on randomly generated subpopulations from the histogram in each window. The calculated uncertainty for the free energy is smaller than 0.22 kcal/mol along the pathway.

Finally, we simulated here for a total of 12 μs, which is longer than or similar to the time scales used to simulate large protein conformational transitions using umbrella sampling with Hamiltonian replica exchange (see for example Moradi and Tajkhorshid, 2013; Ostmeyer, et al., Nature, 501: 121–124; Vukovic, et al., JACS, 2016, 138: 4069−4078; Li, et al., PNAS, 2017, 114: 11145-11150). Together with the relatively small uncertainty, we felt that we had simulated a sufficient long time to be confident about our free energy estimates.

We also checked the behavior of a single replica as the reviewer suggested. In Author response image 1, we plot the trajectories for four different replicas. Two of the four replicas fully covered the reaction coordinate while the other two covered large regions of the reaction coordinate. The free energy profile is computed not just from a single replica, but many. The advantage of umbrella sampling is that by using many replicas (in our case 120), one can sample easily the high-energy regions and guarantee overlaps between neighboring windows. The overlaps and small error estimates have been used in other papers to justify the free energy profiles (see Figure S12 in Moradi and Tajkhorshid, 2013; Figure 7 in Jiang, et al., 2012; Figure S3 in Vukovic, et al., JACS, 2016, 138: 4069−4078). It is not necessary to produce a single trajectory that has “round-trip” behaviors as is the case with simulated tempering methods.

**Author response image 1. respfig1:** Analysis of the reliability of the Hamiltonian replica exchange simulations. (a) Histograms for the umbrella windows near the transition state TS. (**b**) Time evolution of four trajectory samples in the simulations. The closed state and tilted state are labeled at the top and bottom, respectively.

3) More details should be presented in the main text concerning the retrospective analysis of the FRET states, since this analysis is not commonly performed and it is one of the interesting aspects of this work.

We revised the subsection “Analysis of single-molecule data” in Materials and methods and provided more details of the analysis. We also provided more details of the analysis for the dwell time distributions at different ATP concentrations in response to question 6.

4) A strong agreement between the computed FRET distributions for the "tilted" state and the experimental FRET distributions for the re-zipping state are consistent with the authors' proposal that the UvrD conformation during re-zipping resembles the "tilted" state. In a different part of the text, the authors also compare the free energies of the unwinding and re-zipping conformations, which they expect to be similar because UvrD spends equal amount of time in the two conformations. Since the energy of the tilted state is higher than the closed state (the authors even refer to it as a metastable state), it is unlikely to represent the re-zipping state. In fact, the authors introduce a distinct re-zipping state in the Discussion and in Figure 8A, but its structure is not discussed. Please reconcile the two arguments.

The reviewer raised a good point. In the schematic of Figure 8A and B, we show a rezipping state, which has a similar protein conformation as the tilted state but has the bound ssDNA switched from one strand to the other. We believe that the protein conformation does not change much when comparing the tilted state to the rezipping state because the tilted state has a simulated FRET distribution very similar to the experimentally measured FRET distribution of UvrD in the rezipping state.

To explore the structure of the rezipping state further, we built a rezipping structure starting from the tilted conformation (see Author response image 2). Its construction was based on the following considerations: (1) the 2B domain maintains its contacts with the dsDNA while the ssDNA binding domains (1A and 2A) disassociate from one ssDNA strand and binds to the other ssDNA strand. Otherwise, the entire protein would dissociate from DNA. (2) The interaction configuration between ssDNA and the motor domains 1A-2A must remain the same after strand switching. The motor domains move from 3’ to 5’ on the ssDNA in both the unwinding and rezipping modes.

With these considerations, we created a structural model in which we repositioned the UvrD-bound ssDNA segment from the 3’ ssDNA tail (strand A) to the 5’ ssDNA tail (strand B) of the junction in the tilted state. This was achieved by attaching the 5’ terminus of strand B to the 3’ terminus of strand A, and by cutting the ssDNA (strand A) at the junction position. We then equilibrated the modeled system in a water box with 55 mM NaCl for 100 ns. The equilibrated system was stable and had a rmsd around 3Å from the tilted state (see Figure 8—figure supplement 2). The newly obtained state is our prediction of the “rezipping” state (Author response image 2). Such a structure satisfies the prior considerations: (1) The interaction configuration between 2B and dsDNA remains the same between the tilted state and the rezipping state. Note that during our simulations of the closed to tilted transition, the 2B domain changed its contact from one strand of dsDNA to the other (4 bp shift, from the red strand A to the blue strand B in Figure 6). (2) The ATPase domains 1A-2A are in the correct orientation along the ssDNA (3’ to 5’), pointing away from the junction. Such a conformation enables UvrD to translocate along ssDNA, allowing the duplex to rezip behind it. The metastable tilted state has a higher energy than the closed state. While the rezipping state shares the same protein conformational state with the tilted state, the switch of ssDNA strand would make the energy of the rezipping state similar to the closed state.

We have added a new panel to Figure 8 for the newly modeled structure in the main text. We also have added a subsection “DNA-UvrD conformation at the rezipping state” in Discussion and a subsection “Modeling of the rezipping-state structure” in the end of Materials and methods. Finally in examining the rezipping structure, we found a small loop forming between the dsDNA junction and the ssDNA-1A binding site. A similar feature was proposed by a translocation model of PcrA helicase in Park, et al. (2010), which the authors suggested that PcrA can extrude a ssDNA loop while it attaches to dsDNA and translocates the 5’ ssDNA tail in an open conformation.

**Author response image 2. respfig2:** A structural model for the rezipping state. (**a**) The structure of the tilted conformation obtained from the free energy simulations. Strands A and B of the dsDNA are shown in red and blue, respectively. (**b**) A structure model for the rezipping state after the ssDNA switching happens.

5) The apo state of UvrD was ruled out as a re-zipping configuration due to the clashes with the DNA duplex. Please discuss whether these clashes can be removed by allowing the duplex part of the DNA substrate to move/tilt? Transition from the closed conformation into the tilted state results in a substantial movement of the duplex (obvious from Video 1). Perhaps, further movement of the duplex, or positioning of the motor domains on the 5'-terminated strand (which would be consistent with the re-zipping direction) may allow for the DNA bound apo-like state? Modeling such a structure may be beyond the scope of this paper, but some discussion may be informative.

We concur with the reviewer that the UvrD conformational transitions result in motions of the DNA duplex. We ruled out the apo state as a re-zipping configuration due to the following reasons:

a) We forced UvrD from the closed state to the apo state using biased MD simulations in Figure 1—figure supplement 1C. The biasing force was only added to the 2B domain and the DNA was free to move. There was indeed a substantial movement of the duplex when the protein conformation was close to the targeted apo conformation. However, the 2B domain went back to the closed conformation as soon as we removed the external biasing force (see the rmsd change after 50 ns in Figure 1—figure supplement 1C). This indicated that even though the 2B domain was forced to reach the apo conformation, it was highly unstable in such a state. This is consistent with the unfavorable apo conformation in the free energy landscape in Figure 3 in the main text.

b) We simulated the FRET signal for the apo structure by adding fluorophores to apo UvrD (3LFU). The FRET distribution was quite different from the rezipping-state distribution measured by experiments (see Figure 5—figure supplement 1).

c) We have proposed a rezipping-state structure to answer reviewers’ question 4. We then aligned the apo-state conformation to the new conformation. Geometric clashes between the apo 2B and the rezipping-state DNA could be found (see Author response image 3).

We have revised the corresponding text in Results and Discussion.

**Author response image 3. respfig3:** Aligning the apo structure to the new rezipping-state structure. The orange dotted circles indicate clashes between the apo 2B domain and the dsDNA of the rezipping-state structure.

6) All the structures considered here were ATP free. Please state whether there is a reason to believe that the strand switching takes place in the nucleotide-free state?

We appreciate the reviewer for bringing up this question. The simulation results and experiments suggest the strand switching is mostly driven by Brownian motion and does not require energy from ATP hydrolysis. That the ATP is required for unwinding and re-zipping has been established experimentally (see Comstock et al. Science, 2015). To address the possible effect of ATP on strand switching, we (1) analyzed additional data from the optical tweezers experiments obtained at two different ATP concentrations and (2) also analyzed the x-ray structures of the closed state with and without ATP.

We determined the dwell time distributions for the unwinding and rezipping states from optical tweezers measurements at two different ATP concentrations. Details of the measurement are presented in the revised text in the section “Analysis of single-molecule data” of Materials and methods. The dwell time distributions at 10 μM and 2.5 μM are plotted in Figure 8C in the main text. The equilibrium constants of switching measured for the two concentrations are very similar (both around 1), which suggests that strand switching is likely an ATP-independent process. Since 2.5 μM is smaller than Km measured for UvrD, ATP binding states do not seem to have an effect on strand switching. For consistency, we replaced the old plot for the dwell time distribution – where intervals were selected based on the FRET signal – with new plots where intervals were selected based on the optical tweezer signal. We chose to select intervals by the trap signal because it has a higher time resolution than the FRET signal (about one order of magnitude higher).

Finally, we identified all the UvrD structures with ATP or non-hydrolysable analog molecules bound in the bioinformatics analysis. Then we projected the 12 identified structures onto the first two PCs as red triangles (see Figure 8—figure supplement 1). One can see that the 2B motion between the ATP-substrate bound and empty UvrD in the closed state is small relative to the large closed-tilted conformational change. (The conformational changes of the motor domains induced by ATP binding have been demonstrated in Lee and Yang, 2006.) Therefore, it is not very likely that ATP binding has a noticeable impact on the closed-to-tilted transition.

We have added a paragraph about the ATP binding effects in the Discussion.